# Vaginal pH Value for Clinical Diagnosis and Treatment of Common Vaginitis

**DOI:** 10.3390/diagnostics11111996

**Published:** 2021-10-27

**Authors:** Yen-Pin Lin, Wei-Chun Chen, Chao-Min Cheng, Ching-Ju Shen

**Affiliations:** 1Institute of Biomedical Engineering, National Tsing Hua University, Hsinchu 30013, Taiwan; peggy1240309@gmail.com (Y.-P.L.); lionsmanic@gmail.com (W.-C.C.); chaomin@mx.nthu.edu.tw (C.-M.C.); 2Division of Gynecologic Oncology, Department of Obstetrics and Gynecology, Chang Gung Memorial Hospital at Linkou, College of Medicine, Chang Gung University, Taoyaun 333, Taiwan; 3Department of Obstetrics and Gynecology, Chang Gung Memorial Hospital, Keelung 204, Taiwan; 4Department of Obstetrics and Gynecology, Kaohsiung Medical University Hospital, Kaohsiung 807, Taiwan

**Keywords:** vaginal pH, vaginitis, diagnosis, treatment

## Abstract

In modern society, 75% of all women worldwide have had vaginitis at least once in their lives. The vagina has a dynamic microbial ecosystem with varying vaginal pH levels. An imbalance in that ecosystem can alter the vaginal pH and tip the scale to the point of causing issues, such as vaginitis, that require medical attention. Although vaginitis is not an incurable disease, it causes discomfort and pain that disrupt women’s daily lives. The most common causes of vaginitis include bacterial vaginosis, trichomoniasis, and vulvovaginal candidiasis. In this review, we discuss the causes, diagnostic methods, and symptoms of different types of vaginitis, the relationship of vaginitis to the prevalence of other diseases, issues associated with recurrent vaginitis and the immune system, and a variety of effective available treatments. In our article, we summarize the relationship of pH with the vaginal ecosystem, discuss the associated factors of vaginal pH, and finally introduce the different available vaginal pH self-test products.

## 1. Introduction

The vagina serves as an outside-communicating channel with the functions of draining menstruation and childbirth delivery. The vagina normally has unique flora that sustains the internal physical and chemical environment. The presence of normal flora relies on maintenance of various components of the ecosystem, which is in dynamic equilibrium [1]. Based on several published articles, the normal vaginal pH for women of childbearing age ranges from 3.8 to 5.0, which is moderately acidic [2,3]. The normal vagina is covered by a thin layer of transparent liquid, commonly known as vaginal fluid. Many factors may lead to changes or imbalances in the vaginal pH value, including vaginal infections, aging, sexual activity, and vaginal douching [4].

The common vaginal microbiome, the *Lactobacilli* species, can produce acidic pH and bacteriocins to kill other bacteria in the vagina. *Lactobacilli* can produce an acidic environment in the vagina, which is designed to protect women from sexually transmitted pathogens and opportunistic infections [5]. If these normal flora such as *Lactobacilli* are absent or significantly reduced, the vaginal ecosystem will become imbalanced, and other microorganism or bacteria inside vagina may become overgrown, leading to vaginitis. Moreover, Ravel et al. first identified the five community state types (CSTs) in 2011 [6], providing a powerful scheme to classify the status of the human vaginal microbial community (HVMC). These communities are divided into five groups: four groups are dominated by *Lactobacillus iners, L. crispatus, L. gasseri*, or *L. jensenii*, and the fifth has a lower proportion of lactic acid bacteria and a higher proportion of strictly anaerobic organisms.

Vaginitis has different types, including bacterial vaginosis (BV), vaginal candidiasis, trichomoniasis, and aerobic vaginitis [7]. This review focuses on the first three common types of vaginitis. Under the current standard, the diagnosis of vaginitis depends on criteria based on several clinical presentations. For example, the diagnosis of bacterial vaginosis has been based on Amsel Criteria in clinical [1] routine since 1983. Following the Amsel Criteria, bacterial vaginosis is diagnosed by the presence of three out of four conditions, including homogenously milky vaginal discharge, vaginal pH over 4.5, positive KOH whiff test, and 20% at least of clue cells under wet-mount test by microscope [8]. The Nugent Score is a scoring system that calculates the relative number of bacterial morphologies under Gram-stained vaginal discharge smears to diagnose bacterial vaginosis [9]. For both abovementioned diagnostic criteria, the vaginal fluid pH is a useful and unique marker for vaginitis. Abnormal pH values increase the possibility of vaginitis, and the measurement of vaginal pH has been used for initial screening [10]. From previous studies, a vaginal fluid pH value of 4–4.5 or less signifies the absence of vaginitis, whereas a pH value of more than 4.5 denotes vaginitis and bacterial vaginosis (BV) [11]. However, with a trichomonas vaginalis infection, the pH value may be increased to 6.5 or more [12].

There are various signs and symptoms of vaginitis that hint at different types of vaginitis with further corresponding treatments in clinical routine. In previous research, a combination test of vaginal fluid pH value with symptomology was shown to diagnostic sensitivity [13]. Table 1 and Table 2 demonstrate the associated symptoms, signs, risk factors, and treatments of different types of vaginitis [12,13,14,15,16,17]. From this article, we review the relationship among vaginitis, vaginal fluid pH, and the associated immune system. In the closing discussion, we also review the use of commercially available vaginal pH testing products. These vaginal fluid pH test products can serve as self-test tools used at home by patients themselves, especially those with suspected symptoms of vaginitis, such as unusual odor, itching, burning pain, or abnormal vaginal fluid. We hope that this study can increase women’s attention to vaginitis and encourage women to seek treatment as early as possible.

## 2. The Role of Normal Vaginal pH

The pH level associated with the vagina is called the “vaginal pH value,“ and it plays a valuable role in determining vaginal health. The acidic and/or alkaline state are determined by the scale of hydrogen ion activity and measured with the pH value. The naturally neutral pH is equal to 7, but the normal vaginal pH ranges between 3.8 and 5.0, which is moderately acidic [2]. A lower pH value (more acidic) in the vagina than the blood or interstitial fluids can protect vaginal mucosa from pathogenic organisms [4]. The vaginal pH can be affected by overall health conditions, including age, vaginal hydration status, daily diet, and safe intercourse. The vaginal pH value is age-dependent. The normal vaginal pH value for a woman of reproductive age ranges from 4.0 to 4.5, but the value may be slightly higher than 4.5 among premenarchal and postmenopausal women [12].

The vaginal pH value clearly plays an important role in vaginal health, but it is important to note that maintaining a healthy vaginal pH is characterized by the metabolism of *Lactobacillus acidophilus* and other endogenous flora, estrogen, glycogen, and existing flora and pathogens. There is a dynamic relationship between the by-products [16]. Vaginal microorganisms are the primary stabilizers of the vaginal ecosystem. Of those microorganisms, *Lactobacillus acidophilus* is the primary player. This particular microorganism can ferment glycogen derived from the decay of eutrophic vagina mucosa into lactic acid and subsequently release hydrogen ions [18]. The result of this metabolism is an acidic pH of 4–4.5, and the resulting acidic vaginal environment provides a protective effect. It creates a barrier that prevents unhealthy microbiome from multiplying too quickly and causing infection. The imbalance in this ecosystem can cause an unusual vaginal pH and may be used to determine the presence of bacterial pathogens as well as menopausal status [19]. In addition, studies have confirmed that an increase in vaginal pH may lead to bacterial vaginosis (BV) and spontaneous preterm deliveries (PD) in pregnant women [20]. Based on the above research, we know that vaginal pH value has a profound impact on women’s lives. Monitoring that pH level, even with self-testing, can be used to effectively manage and prevent infection.

### Factors That Cause Imbalances in the Vaginal pH

In a woman’s daily life, there are many factors that can cause normal vaginal pH value to become unbalanced, such as unprotected sex, taking antibiotics, vaginal douching, and variations in the menstrual cycle.

Unprotected sexual behavior can lead to an unbalanced vaginal pH [21]. The semen is relatively alkaline, with a pH value of approximately 8.0. and can alter the vaginal pH during unprotected intercourse. Semen can trigger the growth of bacteria that can act as a physiological buffer [22]. Thus, unprotected sex can significantly alter the vaginal pH so that it remains elevated even after 10–14 h [23]. This alteration leaves the vagina less protected against infection.

Antibiotics can inhibit bacterial growth or kill bacteria to treat bacterial infections [24]. In clinical routine, antibiotics are frequently used to treat vaginitis [24,25]. Unfortunately, the antibiotics that kill harmful bacteria also kill good bacteria that maintain a healthy, more acidic vaginal pH value. However, for patients with severe symptoms, antibiotics are still necessary. Antibiotics can rapidly alter the vaginal microbiome within a few hours [26].

Under normal circumstances, the vagina has a self-cleaning function. It does not require any special procedures or solutions beyond normal bathing with clean water. Excessive cleaning or douching of the vagina can not only rinse away vaginal secretion, but can also create an unbalanced vaginal flora with an abnormal vaginal pH environment. Such impaired ecosystem of vagina can cause adverse effects, including BV, pelvic inflammatory disease (PID), pregnancy complications, and even cervical cancer [27,28,29,30]. Thus, the risks of vaginal lavage are far greater than the benefits.

Women’s menstrual cycles are strictly controlled by endocrine, autocrine, and paracrine factors that modulate the endometrial remodeling and regulate the follicular development, ovulation, and luteinization of the ovary [31]. During menstruation, a large amount of menstrual blood flows through the vagina and is absorbed into a tampon or pad and sits in place. The menstrual blood is slightly alkaline and can cause the vaginal pH to rise. Menstrual cycle disorders caused by hormonal imbalance, in addition to the abovementioned abnormal menstrual blood, will also cause vaginal mucosal disorders, which, in turn, affects the microbial microenvironment and causes an increase in vaginitis [32]. For women with a normal, active menstruation cycle, the vaginal pH is typically between 3.8 and 5.0 [3]. Abnormal menstrual cycles are a common feature of puberty. Their existence is related to an increased risk of abnormal pH. The subsequently relatively high vaginal pH may also result in susceptibility to BV [33].

## 3. Common Vaginitis

Vaginitis is a common disorder among women of varying ages, and most women have at least one episode of vaginitis during their lives [34]. Vaginitis occurs because of the introduction of pathogens or changes in the vaginal environment that spread pathogens and change the vaginal flora. Characteristic symptoms, including discharge, odor, itching, irritation, and burning [35], produce discomfort or cause other vaginal complications. These symptoms are related to abnormal vaginal flora [12]. Vulvovaginal complaints are one of the most common reasons for women to seek medical advice [36]. Vaginitis is caused by bacterial vaginosis, vulvovaginal candidiasis, or trichomoniasis [16].

Among all vaginitis cases, between 40% and 50% cases are caused by bacterial vaginosis, between 20% and 25% are caused by vulvovaginal candidiasis, and between 15% and 20% are caused by trichomoniasis. Non-communicable causes, including irritation, allergic, and atrophic and inflammatory vaginitis, are rare and account for between 5% and 10% of all vaginitis cases [14]. The relative symptoms, signs, and risks are organized in Table 1. The differential diagnosis of different types of vaginitis is difficult by symptoms or signs alone. Women with vulvovaginal candidiasis can even present a normal or acidic vaginal pH [14]. Moreover, an inefficacious treatment with poor response may come after an inaccurate diagnosis with further potential sequelae, such as pelvic inflammatory disease [37,38].

Bacterial vaginosis is currently the most common cause of vaginitis. It can be considered a kind of malnutrition that results in the reproduction of anaerobic bacteria and the disappearance of protective *Lactobacillus*, leading to an imbalance in the vaginal flora [39]. This infection is caused by proliferation of several organisms, including *Gardnerella vaginalis*, the *Mobiluncus* species, *Mycoplasma hominis*, and the *Peptostreptococcus* species [40]. Bacterial vaginosis is usually diagnosed with the Amsel criteria and Gram staining [41]. In patients with BV, amines produced by anaerobic bacteria can produce a “fishy” odor, which can predict bacterial vaginosis [42,43]. Bacterial vaginosis may have sequelae similar to pelvic inflammatory disease (PID) and tubal infertility [44,45]. Previous studies have even reported a high prevalence of BV in the non-fallopian tube and unexplained infertility cases [46,47]. There is a high prevalence of BV among infertile patients compared to fertile women (45.5% vs. 15.4%). BV can also be found in 37.4% of patients with unexplained infertility and 60.1% of those with polycystic ovarian disease (PCOD) [48]. Moreover, BV treatment may also have positive effects to pregnancy rates in those infertile women [49].

Vulvovaginal candidiasis is the second most common type of vaginal infection [14]. When the vulva is infected by these microorganisms, this disease may be called vulvovaginitis. It is estimated that 50% of all women suffer from vaginal candidiasis at least once, and vulvovaginal candidiasis accounts for more than 25% of all infectious vaginitis [50]. Although there are many kinds of Candida, the infectious agent in 80–90% of patients is *Candida albicans* [51]. The symptoms of Candida vaginitis commonly include itching, vaginal soreness, dyspareunia, and increased vaginal discharge [43], and women infected with this microorganism typically have a normal vaginal pH value. Candida vaginitis is usually treated by the vaginal administration of imidazole or triazole antifungal drugs, or oral fluconazole [52]. Vulvovaginal candidiasis can be simple or complex according to the clinical manifestations, microbiology, host factors, and response to treatment [53]. The treatment classification is presented in Table 2.

Trichomoniasis, the third most common cause of vaginitis, occurs due to the presence of protozoan Trichomonas vaginalis, an active creature with four flagella [2]. Trichomoniasis is the most common sexually transmitted infection, especially among those with multiple sexual partners [54]. This infection can be found in between 30% and 80% of the male sexual partners of infected women [55]. Patients with trichomoniasis usually have nonspecific symptoms, including increased vaginal discharge, irritation, and itching [54]. In addition, diagnosis is difficult, as 20–50% of women with trichomonas have no symptoms [34,56]. Diagnosis by microscopy is more reliable, and other diagnostic indicators include a positive olfactory test and vaginal pH that is greater than 5.4 [43]. One study reported that this infection increases the transmission rate of human immunodeficiency virus (HIV) [57]. Because trichomoniasis is sexually transmitted with high recurrence rate, the Centers for Disease Control (CDC) recommends re-infection testing 3 months after treatment [53].

### 3.1. Effective Vaginitis Treatment

When people know how to diagnose vaginitis and become familiar with the various symptoms, the most important thing is follow-up care. The standard treatment for bacterial vaginosis is metronidazole (Flagyl) (500 mg twice daily for 7 consecutive days) [53]. There are other treatments, such as Metronidazole gel (Metrogel) (one complete, 5 g applicator daily for 5 days), or Clindamycin 2% cream (one complete 5 g applicator at bedtime for 7 days). If people are allergic or intolerant to metronidazole, vaginal clindamycin cream is the first choice. Cochrane’s review of 24 randomized controlled trials (RCT) indicated that clindamycin and metronidazole (Flagyl) have the same efficacy, with between 91% and 92% of treated patients being clinically cured after 2–3 weeks of treatment [58]. Furthermore, the U.S. Food and Drug Administration (FDA) approved a single-dose oral therapy for bacterial vaginosis, Secnidazole (Solosec), in 2018, and subsequently marketed it [59]. A randomized controlled trial showed that this drug demonstrated comparable efficacy to metronidazole [60]. Previously, the treatment of bacterial vaginosis during pregnancy was recommended to prevent premature birth. A further review article found that antibiotic treatment cannot prevent preterm birth in women with symptomatic or asymptomatic bacterial vaginosis [61]. Therefore, the treatment of bacterial vaginosis can relieve symptoms without other obvious adverse events. It should be noted that recurrence of bacterial vaginitis is common. For patients who have experienced multiple rapid relapses, extended treatment is reasonable.

Treatment options for vulvovaginal candidiasis include oral preparations and topical preparations [62]. Treatment is aimed at alleviating symptoms. There are several topical pyrazole formulations and treatment options, as well as a single dose of 150 mg oral fluconazole (Diflucan) and oral fluconazole, recommended by the CDC as the first-line of treatment for vulvovaginal candidiasis [53]. Medications such as clotrimazole, miconazole, terconazole, and butoconazole are still the most prescribed and are typically used for 3–7 days. In oral medications, fluconazole has replaced ketoconazole, because the former has more favorable side effects [63]. According to the different types of Candida present, we can divide infection into simple or complicated vaginal candidiasis. Patients with complicated Candida vaginitis require more aggressive treatment. Most recurrent vaginal candidiasis is associated with *Candida albicans*, and it had been proven helpful to administer intensive treatment with fluconazole for 7 to 14 days (fluconazole 150 mg once every three days for a total of three doses). Further fluconazole treatment for 6 months (150 mg per week) can relieve symptoms in 1 year [64].

Trichomonas vaginalis is a human protistan parasite and the most common non-viral sexually transmitted disease worldwide. In these cases, the cause of vaginitis is *Trichomonas vaginalis*, which can also infect the prostate and urethra in men [65]. Currently, common treatments for trichomoniasis include 5-nitroimidazole drugs through oral and parenteral routes. In the United States, only metronidazole and tinidazole have been authorized by the U.S. Food and Drug Administration (FDA) and are available for the treatment of trichomoniasis [66]. In 90% of all cases, a single oral or long-term administration of nitroimidazole drugs can cure this parasitic disease [67].

Gastrointestinal discomfort is the most common side effect of metronidazole, and it is usually mild and tolerable, but may be severe under high doses for those with refractory trichomoniasis [68]. Tinidazole is a nitroimidazole that was introduced in 1969 to treat vaginal infections caused by Trichomonas bacteria. The therapeutic dose of tinidazole is lower than that of metronidazole, and it has milder and fewer side effects [69]. The cure rate of parasites in vaginal nitroimidazole cream is less than 50%. In addition, better efficacy has been found in those with combined oral and vaginal treatment than oral treatment alone from several randomized controlled trials [67,70]. It is suggested that sexual partners should receive treatment at the same time, and that both partners should avoid sexual intercourse until they have completed treatment without further symptoms.

The administration of oral lactic acid bacteria to alter the vaginal microbiota seems to be effective. The Nugent score has shown improvement when patients ingest a mixture of 108 L. fermentum 57A, L. plantarum 57B, and L. gasseri 57C daily for 60 days. These bacteria can settle in the rectum and vagina from day 20 to day 70 and reduce the vaginal pH [71]. Another study showed that oral administration of Lactobacillus acidophilus, Lactobacillus rhamnosus GR-1, and Lactobacillus fermentum RC-14 at a dose of at least 108 CFU/day per day can provide therapeutic effects [72]. When intestinal bacteria are kept in balance and healthy, the vaginal microbiota may also align and improve. Since Lactobacilli can create a defense system against dysbiosis and infections within the vagina, a daily diet with addition of Lactobacillus can also improve vaginal flora and reduce the incidence of vaginitis. Lactobacillus supplements can maintain a weakly acidic vaginal environment, reduce urogenital infections, and form a natural protective barrier.

### 3.2. Recurrent Vaginal Candidiasis and The Immune System

Recurrent vaginal candidiasis is not uncommon [73]. Although the medical treatment of vaginal candidiasis can relieve symptoms, the infection usually relapses after therapeutic drugs are terminated. Temporary and local suppression of cell-mediated immunity can lead to recurrent vaginitis [74]. Lymphocytes from many women with this disease have shown a reduced in vitro proliferative response to *Candida albicans*. It seems that the inhibitory effect is due to the increased production of prostaglandin E 2 in the patient’s macrophages [75], thereby inhibiting the production of interleukin 2, which prevents the proliferation of lymphocytes. When the lymphocyte response is impaired, *Candida albicans* can easily proliferate and initiate clinical infection.

Recurrent vaginal candidiasis may be related to the resistance of the antibiotic treatment, and progressed symptoms were frequently seen under four or more episodes relapsed per year [76]. Although long-term maintenance treatment with fluconazole can treat relapsed vaginal candidiasis, it is difficult to use such approach for extended periods of time [64]. Repeated treatments may cause drug resistance, change the range of pathogenic Candida species, and increase the incidence of non-communicable diseases.

In 2005, Wozniak et al. [77] demonstrated that cell-mediated immunity (CMI) acted as the main host defense mechanism against most Candida infections, but CMI provided no protection against systemic or local Candida infection. Specific CMI or antibodies against vaginal candidiasis have been identified. There is evidence of immunomodulation in vaginal tissues, and the more profound protective Th1 type response may be suppressed. Wozniak’s study evaluated the possibilities for overcoming immune regulation and enhancing protection against vaginal candidiasis using immunotherapy and gene therapy in mouse models. Adenomyosis can induce tropism of vaginal tissue, and it makes the application of immunotherapy tool possible to treat venereal diseases or other vaginal diseases requiring the control of local immune responses. However, the cytokine immunotherapy and adenovirus gene therapy still cannot successfully improve the protective effects for diseases, including vaginal candidiasis [78].

In 2015, Bernalis et al. used an animal model of vaginal candidiasis to characterize the mechanism that induces mucosal immunity against *Candida albicans*, and addressed the related interaction between innate immunity and adaptive immunity [76]. Flavia’s study reported that Th1 protective immunity could induce cell-mediated immunity (CMI) and antibody (Abs)-mediated immunity. Overall, their data provide clear evidence that it is possible to prevent vaginal infections by *Candida albicans* through active intravaginal immunization with aspartyl proteases expressed as recombinant proteins. These results indicate that a vaccine composed of virions and secreted aspartyl protease 2 (Sap2) (PEV7) may prove beneficial for the treatment of recurrent vaginal candidiasis. The integration of the two abovementioned studies could lead to a Candida vaginitis vaccine that might more effectively relieve suffering and treat the problem of recurrent vaginitis via immunotherapy.

### 3.3. Infection Diagnosis with Vaginal pH

The normal vaginal pH theoretically ranges from 3.8 to 5.0, and it may be affected by vaginitis, especially under changes of the vaginal ecosystem by topical medication or unprotected sex. Therefore, we can consider the measurement of vaginal pH to be a useful screening tool for determining vaginitis. Here, we review the relationship of vaginitis to the vaginal pH value.

The replacement of normal vaginal lactobacilli by anaerobic bacteria raises the vaginal pH above 4.5 and leads to the onset of bacterial vaginosis. When infected with trichomonas, the vaginal pH is usually greater than 5.4. Women infected with Candida vaginitis typically have a normal vaginal pH [36], but higher vaginal pH values have also been reported [13]. In 1985, Hanna et al. investigated the relationship between vaginal pH and microbiological status in vaginitis. They found that vaginal microbial status was changed while the vaginal pH was in the range of 5.0–5.5 or 6.0–7.5, supporting the view that vaginal pH increases during vaginal infection [79].

J. Thinkhamrop et al. [80] assessed vaginitis via the medical history, comprehensive physical examination, and vaginal examination of outpatient gynecological patients in Obstetrics and Gynecology Clinic of Srinagar Linde Hospital from 1 May to 31 July 1997. They also collected specimens for microbiological examination and pH measurement. Their experimental results demonstrated that the diagnostic performance of a vaginal fluid pH test in combination with clinical symptoms and signs as a screening test for vaginitis was more reliable than using a pH test alone. When a pH test was used as the only means of diagnosing BV, the sensitivity was higher than that for all other types of vaginitis. In summary, the sensitivity of a vaginal fluid pH test for determining infectious vaginitis is approximately 50%. The sensitivity increases to 73% when applied for BV vaginitis only. The sensitivity to vaginitis due to fungal infection is only 22%. When the vaginal fluid pH test is combined with other medical diagnosis, the sensitivity is relatively improved and increases to 67.5%.

In addition, variations in the vaginal fluid pH may be caused by unprotected sex, the use of antibiotics, vaginal douching, or menstrual cycle changes. The sensitivity of using only a pH test to diagnose vaginitis is 66% according to previous research [81]. Using a pH test combined with clinical symptoms to screen for BV can increase the test sensitivity to 81.3% (95% CI: 69.2–89.5) (Table 3). The reason why the pH test is more sensitive to BV is that the bacteria occupy the ecology of the vagina, which reduces the number of lactobacilli and the acidic substance secreted by them, thus increasing vaginal pH. Therefore, vaginal pH tests can serve as a tool for the clinical detection of BV. Although the sensitivity is limited by using vaginal pH test alone, it is still a very convenient tool for women to use by themselves to monitor their vaginal health, especially while experiencing vaginitis symptoms.

## 4. Vaginal pH Test Products

If women experience abnormal vaginal symptoms such as itching, burning, unpleasant vaginal odor, or abnormal vaginal discharge, then they may need to test their vaginal pH. However, the U.S. Food and Drug Administration (FDA) notes that women should understand that at-home tests will not help diagnose HIV, chlamydia, herpes, gonorrhea, syphilis, or group B streptococcus [82]. It is clear that clinicians already use vaginal pH testing to help diagnose the causes of vaginitis [8,19]. With the increasing demand for self-diagnosis and self-treatment, pH tests allow women to self-manage some of their health care. This review details the use of such self-test tools for screening purposes and recommends their use as a rapid, simple, and effective early screening tool.

The Hygeia Touch Self-Testing Kit for Vaginal Infection [83] (Hygeia Touch Inc., Taipei, Taiwan; MHW Medical Device Manufacturing No. 006714) uses a vaginal applicator that includes a bromocresol green pH indicator embedded into a biocompatible grip. This test stick is assembled from a pH test paper and a biocompatible plastic stick. This tool is inserted into the vagina to collect a sample, and then allowed to rest for 1 min to allow the secretions to react with the pH test paper. Colorimetric results indicate pH level and can help distinguish the cause of infection, i.e., *Candida albicans*, bacteria, or trichomonas. Furthermore, this device has been registered with the U.S. FDA.

Another product on the market, the Biosynex Exacto 3 vaginal infection test [84], has demonstrated reliability and accuracy (90%). It is easy to use, can produce immediate results, and is suitable for the preliminary diagnosis of vaginal infection. This product provides straightforward vaginal contact sampling and presents rapid, easy-to-interpret, and simple color-coded results. There are several other commercially available vaginitis self-diagnostic devices summarized in Table 4.

Among all available products, The Hygeia Touch Self-Testing Kit for Vaginal Infection has a distinct advantage. The double-layer protection design prevents the test paper from falling off and not directly touching the skin, and the elastic baffle design ensures that the depth of insertion is not overly deep. The use of flexible, medical-grade plastic makes it comfortable and safe for use. The appropriate use of these devices allows patients to monitor the disease course and select the correct over-the-counter (OTC) antifungal drugs, which the FDA approved in 1990. These products can be used by patients themselves, enhance the caution of vaginal health, and facilitate the maintenance of vaginal pH and overall health.

## 5. Summary

Vaginal care is a serious issue for women. Variations in vaginal health may have widespread effects, and vaginal pH plays a significant role. Therefore, pH can be very useful for diagnosing, monitoring, and treating conditions associated with vaginitis. We outline the affecting factors of vaginal pH and addressed the different types, symptoms, risks, and effective treatments for vaginitis. The recurrence of vaginitis also appears to be related to immune system factors. Vaginal pH self-test products can be used to monitor the vaginal pH to detect vaginitis, especially when paired with other diagnostic measures. Such tool can serve as not only a point-of-care test in clinics but also a self-test at home, and it can enhance the motivation of vaginal health inspection and further facilitate vaginal health.

## Figures and Tables

**Table 1 diagnostics-11-01996-t001:** Symptoms and signs of vaginitis (Information from [12,13,14,15,16,17]).

Diagnosis	Etiology	Symptoms	Signs	Risks	pH Value
**Bacterial vaginosis**	Anaerobic bacteria (Prevotella, Mobiluncus, Gardnerella vaginalis, Ureaplasma, Mycoplasma)	Fishy odor; malodorous; homogenous; clear, white, or gray discharge that may worsen after intercourse; pelvic discomfort may be present.	No inflammation.	Increased risk of HIV, gonorrhea, chlamydia, and herpes infections.	greater than 4.5
**Vulvovaginal** **candidiasis**	Candida albicans, Candida krusei, Candida glabrata	No odor; white, thick, cheesy, or curdy discharge; vulvar itching or burning.	Signs of inflammation; Vulvar erythema and edema.	vulvodynia	4.0
**Trichomoniasis**	Trichomonas vaginalis	Green or yellow, frothy discharge; foul odor; pain with sexual intercourse, vaginal soreness, dysuria.	Signs of inflammation, “strawberry cervix”; Vestibular erythema may be present.	Increased risk of HIV infection Increased risk of preterm labor. Should be screened for other sexually transmitted infections.	5.0~6.0

**Table 2 diagnostics-11-01996-t002:** Treatment for the most common vaginitis (Information from [13,14,15,36]).

	Treatment	
	Initial Regimen	Alternative Regimen
**Bacterial vaginosis**	Metronidazole (Flagyl), 500 mg orally twice daily for seven days Or Metronidazole 0.75% gel (Metrogel), one full applicator (5 g) intravaginally daily for five days Or Clindamycin 2% cream, one full applicator (5 g) intravaginally at bedtime for seven days	Tinidazole (Tindamax), 2 g orally once daily for two days or Tinidazole, 1 g orally once daily for five days Or Clindamycin, 300 mg orally twice daily for seven days or Clindamycin (Cleocin Ovules), 100 mg intra-vaginally at bedtime for three days
**Vulvovaginal candidiasis**	Topical azole therapy or Fluconazole (Diflucan), 150 mg orally, single dose	Medical-grade honey (MGH)
**Trichomoniasis**	Metronidazole, 2 g orally, single or divided dose on the same day Or Tinidazole, 2 g orally, single dose	Metronidazole, 500 mg orally twice daily for seven days

**Table 3 diagnostics-11-01996-t003:** Diagnostic performance of vaginal fluid pH and other tests in screening for vaginitis 1–3 and for BV 4–6 (Information from [80]).

Test	Sensitivity (95% C.I.)	Specificity (95% C.I.)	Accuracy (95%C.I.)
**pH screen for vaginitis**	49.7% (42.6–56.9)	75.5% (69.0–81.0)	63.0% (58.1–67.7)
**pH + clinical symptoms and signs to screen for vaginitis**	67.5% (60.4–73.9)	62.0% (55.0–68.6)	64.7% (59.8–69.4)
**Clinical symptoms and signs to screen for vaginitis**	38.6% (31.9–45.7)	77.7% (71.5–82.9)	59.0% (54.1–63.7)
**pH screen for BV vaginitis**	73.4% (60.7–83.3)	70.1% (64.9–74.8)	70.6% (65.9–75.0)
**pH + clinical symptoms and signs to screen for BV vaginitis**	81.3% (69.2–89.5)	53.1% (47.6–58.5)	57.5% (52.6–62.4)
**Clinical symptoms and signs to screen for BV vaginitis**	39.1% (27.4–52.1)	71.5% (66.5–76.1)	66.6% (61.9–71.1)

**Table 4 diagnostics-11-01996-t004:** Summary of commercially available vaginal self-test products.

Brand	Hygeia Touch	Biosynex	FloriSense	Monistat
**Appearance**	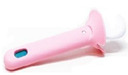	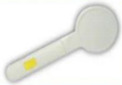	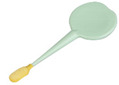	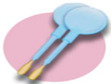
**Accuracy**	88%	90%	92%	92%
**Test/pack**	1	3	2	2
**Advantage**	The double-layer structure is optimized for product safety and ease of use.	Because there are 3 tests in each box, there is an advantage in quantity.	Reliable accuracy is over 90% and product is easy to read.	Effective for the diagnosis of yeast-based vaginal infections

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
