# Peer review of "Vaginal pH Value for Clinical Diagnosis and Treatment of Common Vaginitis"

_diagnostics, 2021, doi:10.3390/diagnostics11111996_

Round 1
Reviewer 1 Report
Dear Authors,
Thank you for providing the revised version. I have no further comments.
Kind regards,
Author Response
Thank you for your note and your response to our manuscript, “Vaginal pH value for Clinical
Diagnosis and Treatment of Common Vaginitis” Again, we very much appreciate and are very
encouraged by the reviewers’ comments regarding our manuscript. Below, we have addressed the reviewers’ comments point-by-point and have attached a revised manuscript with the associated changes, which are highlighted in yellow. Please let us know if there is anything else that we should address with respect to this manuscript. Thank you for the consideration, and we look forward to
hearing from you.
Reviewer 2 Report
This review is easy to follow and well-structured. It builds up the case for home-pH-testing very nicely, before delving into it. Section 3.2, however, seems outside the scope of this review on vaginal pH. Either make this connection explicit, or remove this section.
In general, while I respect your choice of focusing on vulvovaginal candidiasis, bacterial vaginosis and trichomoniasis, it would be positive to include at least cursory information on aerobic vaginitis, including its most common infectious agents, symptoms and associated vaginal pH. This review might be a good starting point: https://pubmed.ncbi.nlm.nih.gov/28502874/
In addition, I found on direct quote in the text without quotation marks. While I haven't found any other such occurrences, I encourage the authors to carefully assess their text and whether they may be plagiarizing without intent.
Finally, section 4 is missing a reflexion on the role of self-testing, other than "women want it". Can this lead to better health outcomes? Could it lead to worse outcomes, in case it leads to self-medication or delay in seeking healthcare?
In addition to these general comments, I have several minor issues with the manuscript, listed below.
Throughout the text: replace flora with microbiome
Throughout the text: italicize genus and species names
Throughout the text: please remove all highlights and editing marks that are not relevant for reviewers.
line 38: "can produce acidic pH" - be explicit. Which acids are secreted by Lactobacilli?
line 53: the Nugent criteria are commonly used in research settings, but not in the clinic. Amsel criteria are common in clinic settings.
line 62: vulvovaginal candidiasis usually presents with a vaginal pH in the 4-4.5 range
line 86: what is "premarche"? Did you mean "menarche"?
line 89 - L. acidophilus is not in any way required for maintenance of vaginal pH. L. crispatus, L. gasseri and L. jensenii, amongst others, can adequately maintain vaginal pH. This statement is in direct contradiction to Ravel's CST described earlier in the text.
lines 88-90 are taken almost ipsis literi from reference 16. Rewrite and update it, or at least put the direct citation in quotation marks. Please be careful to put all direct citations in quotation marks.
line 91: define and describe "Döderlein's bacillus", as this is a historical term and not a valid taxonomic classication
line 96 - again, vulvovaginal candidiasis usually presents with regular vaginal pH
lines 103/104 - please remove this line. It is implying that women are too blame for their BV or VVC, while not offering concrete guidance in avoiding these conditions.
line 114: add a citation for the pH of semen
lines 122-124: considering antibiotics are prescription drugs in most countries, is this statement needed?
lines 127-129: add references to these statements
lines 124-135: "endometrial remodelling of the ovary" doesn't make any sense. Please be explicit on what is happening in the ovaries and what is happening in the uterus.
lines 133-141: the menstrual cycle also affects the vaginal lining and its mucus, potentially affecting pH even in the absence of menstrual bleedings. Please discuss these direct hormonal effects as well.
lines 149-153: I don't understand why the use of NGS technologies is placed in this paragraph about frequency of vaginitis. This is a research tool and should be discussed in an appropriate section, or omitted entirely.
lines 169-171: if the women are asymptomatic, they shouldn't present with any symptoms, including malodour
lines 177-178: present a reference to this statement
lines 163-178: BV may not be directly linked to infertility, and rather be driven by the same hormonal imbalance. Discuss this issue.
lines 206-212: these lines are written in second person ("you"), and this is extra confusing considering in one sentence "you" are a healthcare provider, and in the next "you" are a patient. Be explicit about who is meant in each sentence.
lines 240-241: Trichomonas vaginalis, not Trypanosoma
lines 257-268: there are about as many succesful as not succesful studies of oral probiotic supplementation to treat or prevent vaginosis. Your review should be balanced and include some of these unsuccesful studies as well.
line 327: there is no theory requiring vaginal pH to be in the 3.8-5.0 range. These are empirical observations.
line 392: again, do not use the second person "you" if you're refering to patients or women in general
Table 1: VVC is linked to an increased risk of vulvodynia
Table 2: if in the text you describe several alternative treatments for VVC, why is there only one in the table?
Figure 1: please verify and state that this image is reproduced after authorization of the authors or under a CC license.
Author Response
Thank you for your professional and valuable comments, please see the attachment.

Round 2
Reviewer 2 Report
The authors have addressed all of my earlier concerns. I disagree with the role the authors assign to L. acidophilus, which I find to be based on outdated knowledge, but the authors have been clear with their line of thought.
My one remaining concern is that figure 1 is not appropriately cited, if it's not created by the authors. However, I'll leave this issue up to the editor, as the journal is liable for any potential misuse of data.
This manuscript is a resubmission of an earlier submission. The following is a list of the peer review reports and author responses from that submission.
Round 1
Reviewer 1 Report
The authors performed a review on the value of the vaginal pH for Clinical Diagnosis and Treatment of common vaginitis. The manuscript is well written and easy to understand. However, unfortunately, there are several important limitations.
Major comments
The review is not updated. New concepts and recent advances on the knowledge of vaginal infections are missing. I suggest including in the review the following topics:
- The microbiome.
- The use of next generation sequencing technologies to evaluate vaginal flora.
- The aerobic vaginitis as newly defined clinical entity that is distinct from candidiasis, trichomoniasis and bacterial vaginosis.
- The five community state types (CSTs) to classify the states of human vaginal microbial.
Many references date back to the 1980s or 1990s.
Minor comments
There are some errors in English grammar and spelling.
Author Response
Thank you for these valuable comments. We have discuss ed the next generation sequencing technologies in our revised manuscript (Line 43-48, Page1-2 , as yellow highlighted) highlighted). In addition, since our review
mainly focused on the common vaginal flora including anaerobic bacteria, candida, and trichomanias we have revised our introduction with the updated new flora (Line 304 319, Page 8, as yellow highlighted). We have significantly revised our manuscript with the updated references.
Reviewer 2 Report
I congratulate with the authors for the nice paper.
It is easy to read and provides a summary of what's currently on the market and the rationale behind.
I have no comments
Author Response
Thank you very much for this comment on our manuscript.
Reviewer 3 Report
Dear Authors,
The manuscript entitled "Vaginal pH Value for Clinical Diagnosis and Treatment of Common Vaginitis", is an interesting review addressing a women's common issue, Vaginitis. The authors provide a clinical view of this pathogenic condition. They describe the different types and causes of vaginitis, the signs and symptoms, the diagnosis and the treatment. They analyse the relationship between vaginal pH and Vaginitis, they also analyse the pH tests available as a diagnostic tool.
The manuscript is well written, well-organized. It is clear and concise. In my opinion, the work just needs a few corrections:
- lines 33 and 101 states that the normal vaginal pH is within the range of 3.8 and 4.5 however, lines 74 and 301 reports the normal vaginal pH range within 4.0 and 4.5. In both cases, reference 2 is cited. Please review.
- Figure 1 is not cited in the text, please include it.
- Lines 131-132: vaginal pH between 4.0 and 5.0. What condition(s) is/are different from lines 33, 74, 101 and 301? Please clarify why the pH range is wider in this case than the previous values reported. I assume there is a difference, but it is a bit confusing.
- Table 1 presentation can be improved: i.e. the size of the letters in symptoms and signs is bigger than the rest of the header. Maybe adding a new column considering each vaginitis type show pH change would be of interest.
- line 206: Secnidazole? (according to reference 56, a "c" is missing), please review.
- lines 276 and 280-281: "Karen et al." Please review the citation, according to reference 72 is "Wozniak et al."
- lines 288 and 290: "Flavia et al." Please review the citation, according to reference 71 is "De Bernalis et al."
- Table 3 is not cited in the text, please include it. Additionally, the specificity and accuracy of the vaginal pH and other tests included in the table are not discussed.
- Line 345: please review the sentence: "It I clear that clinicians already use vaginal pH testing to help diagnose the causes of vaginitis".
- Line 359: according to table 4 is exactly 90% of accuracy, not bigger than 90%. Please review.
- Line 362: please correct the sentence: "There areseveral other commercially available [...]"
- In my opinion, there is a lack of recent references: only 6 out of 82 references are from 2016 on. As far as I am concerned, including more recent references will give the manuscript a trendy look.
Kind regards,
Author Response
Thank you very much for these valuable comments. Regarding the comment No. 1 to No. 11, we have made changes in our manuscript (as below list). Regarding the comment No. 12, we have placed a new section, Use Next-Generation Sequencing as a Clinical Diagnostic Tool in Vaginitis (Line 43-48, Page1-2 , as yellow highlighted).
Comment 1
Line 34 & 80 & 322, Page 1& 2 & 8.
Comment 2
Line 77, Page 2.
Comment 3
“For women with a normal, active menstruation cycle, vaginal pH is typically between 4.0 and 5.0 [29].” It’s means that women in menstruation cycle have the vaginal pH around 4.0-5.0 a little above than not being in menstruation cycle.
Comment 4
Page 5.
Comment 5
Line211, Page 5.
Comment 6
Line 281& 286, Page 7.
Comment 7
Line 293, Page 7.
Comment 8
Line 351, Page 9.
Comment 9
Line 365, Page 9.
Comment 10
Line 379, Page 9.
Comment 11
Line 382, Page 10.
Comment 12
We add some from 2015 on paper.
Round 2
Reviewer 1 Report
I thank the authors for their efforts. However, the manuscript was only slightly improved. I confirm that in my opinion this article is of limited interest to the reader.